# Educational Inequalities in COVID-19 Vaccination: A Cross-Sectional Study of the Adult Population in the Lazio Region, Italy

**DOI:** 10.3390/vaccines10030364

**Published:** 2022-02-25

**Authors:** Giulia Cesaroni, Enrico Calandrini, Maria Balducci, Giovanna Cappai, Mirko Di Martino, Chiara Sorge, Emanuele Nicastri, Nera Agabiti, Marina Davoli

**Affiliations:** 1Department of Epidemiology-Regional Health Service, ASL Roma 1, Via Cristoforo Colombo 112, 00147 Rome, Italy; e.calandrini@deplazio.it (E.C.); m.balducci@deplazio.it (M.B.); g.cappai@deplazio.it (G.C.); m.dimartino@deplazio.it (M.D.M.); c.sorge@deplazio.it (C.S.); n.agabiti@deplazio.it (N.A.); m.davoli@deplazio.it (M.D.); 2Clinical Division of Infectious Diseases, National Institute for Infectious Diseases Lazzaro Spallanzani-IRCCS, 00149 Rome, Italy; emanuele.nicastri@inmi.it

**Keywords:** COVID-19, vaccine, socioeconomic factors

## Abstract

Several studies reported socioeconomic inequalities during the COVID-19 pandemic. We aimed at investigating educational inequalities in COVID-19 vaccination on 22 December 2021. We used the cohort of all residents in the Lazio Region, Central Italy, established at the beginning of the pandemic to investigate the effects of COVID-19. The Lazio Region has 5.5 million residents, mostly distributed in the Metropolitan Area of Rome (4.3 million inhabitants). We selected those aged 35 years or more who were alive and still residents on 22 December 2021. The cohort included data on sociodemographic, health characteristics, COVID-19 vaccination (none, partial, or complete), and SARS-CoV-2 infection. We used adjusted logistic regression models to analyze the association between level of education and no vaccination. We investigated 3,186,728 subjects (54% women). By the end of 2021, 88.1% of the population was fully vaccinated, and 10.3% were not vaccinated. There were strong socioeconomic inequalities in not getting vaccinated: compared with those with a university degree, residents with a high school degree had an odds ratio (OR) of 1.29 (95% confidence interval, CI, 1.27–1.30), and subjects with a junior high or primary school attainment had an OR = 1.41 (95% CI: 1.40–1.43). Since a comprehensive vaccination against COVID-19 could help reduce socioeconomic inequalities raised with the pandemic, further efforts in reaching the low socioeconomic strata of the population are crucial.

## 1. Introduction

Although media and scientific editorials addressed COVID-19 as a great leveler at the beginning of the pandemic, commentaries and studies on socioeconomic inequalities during the pandemic soon appeared from several countries, raising solid concerns [1,2]. Even in high-income countries, compared with the advantaged groups of the population, the disadvantaged have a higher risk of SARS-CoV-2 infection [3], higher hospitalizations for COVID-19 [4], increased admissions to an intensive care unit [3], higher mortality rates [5,6,7], and lower probability of being tested for SARS-CoV-2 infection [3].

Social determinants of health have been deeply investigated in Europe and are present in all countries [8]. In the Lazio Region, despite a universal Regional Health Service that aims to provide health care for all, socioeconomic inequalities in health and access to health care have been reported extensively [9,10,11,12,13,14]. Differences between social positions have been reported in life expectancy [9], in all-cause and cause-specific mortality [10], in the incidence and prognosis of disease [11], in hospitalizations [12], and in adherence to evidence-based drug therapies [13]. In addition, during 2020, educational inequalities in SARS-CoV-2 incidence and mortality within 30 days from the onset of infection were analyzed [14], showing strong disparities in the last term of the year. In 2020, the vaccines against COVID-19 were not available yet, and the vaccination campaign started at the end of December 2020. Although a comprehensive vaccination against COVID-19 could help reduce socioeconomic inequalities, several studies investigated the characteristics associated with the willingness to receive the vaccination against COVID-19 [15,16,17,18,19], showing greater hesitancy in the disadvantaged groups of the population. 

In Italy, vaccines were first offered to health care workers, then to the general population. The vaccine administration to the general population followed priority criteria based on individual vulnerability. The first to be vaccinated were residents aged 80 years or more, disabled individuals, or subjects with severe comorbid conditions. The second were those aged 70–79 years. Then the 60- to 69-year-olds, followed by those aged less than 60 years with comorbid conditions or occupations characterized by high levels of exposure (teachers, police, and armed forces). Finally, vaccines were offered to younger groups of the population. Access to vaccination was possible through a general practitioner or individual booking on the Regional Health Service website. In December 2021, all the adult population who wanted a COVID-19 vaccination had the possibility of being fully vaccinated. The present study aimed at evaluating the association between educational level and not getting vaccinated on 22 December 2021 and at identifying specific targets for vaccination campaigns.

## 2. Materials and Methods

### 2.1. Setting and Study Design

We conducted the study in the Lazio Region, one of the 21 Italian regions or autonomous provinces. Located in Central Italy, Lazio has a population of about 5.5 million inhabitants. The population lives primarily in the Metropolitan Area of Rome (4.3 million inhabitants) and Rome, the Italian capital city, with 2.8 million residents. The Regional Health Service, a universal health care system, provides health care to all residents. The health care workers were the first to get COVID-19 vaccination from December 2020, then the entire population from the elderly to the 12-year-olds during 2021. Eventually, in December 2021, all adult residents could get a complete vaccination. Hence, we used a cross-sectional design to investigate socioeconomic inequalities and characteristics associated with not being vaccinated on 22 December 2021.

### 2.2. Sources of Data

We used the cohort of all residents in January 2020 established to investigate COVID-19 determinants and consequences within the project DeteCOVID, funded by the Italian Ministry of Health. The cohort was based on the record-linkage of the Regional Longitudinal Study and the Integrated Surveillance System of SARS-CoV-2 infections. 

The Regional Longitudinal Study is the administrative cohort of residents in the Lazio Region created by linking the Regional Health Information System with the census of the population. The Regional Health Information System contains all health administrative Regional Health Information System databases, including the patient, the hospital discharge, and the mortality registries. The census contains information on individual education. The Lazio Region Longitudinal study, which includes anonymized data, is part of the National Statistical Program and is approved by the Italian Data Protection Authority each year. 

The Integrated Surveillance System of SARS-CoV-2 infections was established in the Lazio Region in February 2020 to collect individual data on all patients with a positive test for SARS-CoV-2, with the date of infection, recovery, and vaccination. Since the last census of the population was conducted in 2011, the information on the individual level of education was available for those living in the Lazio Region on the census reference day (9 October 2011) and can be considered valid for those who at the census reference day had completed the educational process, i.e., 25 years of age for university degrees. Each resident is assigned a unique anonymized identifier that allows record linkage across all databases according to Italian regulations concerning the handling of personal data. The linkage between the Lazio Region Longitudinal Study and the Integrated Surveillance System of SARS-CoV-2 infections was possible within the DeteCOVID project.

### 2.3. Study Population

From the cohort DeteCOVID of all residents, we selected subjects aged 35 years or more on 1 January 2020 who presumably had completed education by the 2011 census. Moreover, we selected those who had information on the presence or absence of chronic comorbid conditions and who were alive in December 2021. Finally, we investigated residents with the information on attained education (88% of the selected adult population). Figure 1 shows the selection process.

### 2.4. Outcome and Other Variables

We created a variable indicating whether the subjects had a complete vaccination, a partial vaccination, or if he/she was not vaccinated at all. As complete vaccination, we intended two doses of Pfizer-BioNTech, Moderna, or AstraZeneca vaccines, one single dose of Johnson and Johnson, or one single dose of any vaccine within a year from SARS-CoV-2 infection. Additionally, as partial vaccination, we considered those who were infected in the six months preceding 22 December 2021 or those who had only one dose on the same date. As additional variables, we considered gender, age (as a continuous variable), place of birth (Italy or abroad), number of chronic conditions at the beginning of the pandemic (0, 1, 2, 3, or more), level of education (high for those with a university degree, medium for those with high school degrees, low for those with junior high or primary school degrees), deprivation index (a composite indicator of the census block of residence based on 2011 census data and divided into quintiles weighted by the population) [20], and previous SARS-CoV-2 infection.

### 2.5. Statistical Analysis

We tabulated the characteristics of included and excluded subjects. We described the population by vaccination status, and we performed logistic regression models to investigate educational inequalities and characteristics associated with not getting vaccinated. First, we performed a model adjusted for age and sex, then a fully adjusted model. Finally, we investigated the possible interaction between sex and variables in their association with no vaccination using the log likelihood ratio test.

## 3. Results

After the selection process, shown in Figure 1, we used a population of 3,186,728 residents with complete information on educational level. Appendix A shows the characteristics of included and not included subjects. Compared with subjects included in the study, those excluded for missing information on educational level were younger, more foreigners (37.5% vs. 8%), healthier (7.4% had three or more chronic conditions vs. 14%), and less vaccinated (70% vs. 88%).

On 22 December 2021, 88.1% of our study population was fully vaccinated, but 10.3% had not received a single dose of vaccine against COVID-19.

Table 1 shows the characteristics of the study population according to the vaccination status. The proportion of not vaccinated was slightly higher in women than in men, and it was much higher in foreigners (25.4%) than in Italians (9.0%); it increases with decreasing level of education or with increasing census block level of deprivation, and it decreased with an increase in the number of chronic conditions.

Table 2 shows the results of logistic regression models. Since there was no evidence of an effect modification by sex and socioeconomic indicators, we report the overall results. The age-adjusted odds ratio showed that women were 8% more likely than men not to get vaccinated. Foreigners, who theoretically had the same access as Italians to vaccination because legal residents are assisted by the RHS, were three times more likely than Italians not to get the COVID-19 vaccine. Once age and sex were taken into account, residents with low education were 40% more likely than the highly educated not to get vaccinated. The same pattern was obtained across the quintiles of the deprivation index: compared with residents in the least deprived areas, the subjects living in the most deprived census blocks had an OR = 1.31 (95% CI: 1.29–1.32) of no vaccination, with a statistically significant trend across quintiles of deprivation (*p* < 0.001). Worse health conditions (number of pre-existing chronic conditions) were associated with vaccination against COVID-19. Once other confounders in addition to age and sex were taken into account (OR2), the associations remained stable.

## 4. Discussion

Our population had a high vaccination rate (88%), but we found solid inequalities in the COVID-19 vaccination. Low-educated residents had a 40% higher probability of not getting vaccinated than those with a university degree. The results were confirmed using a small-area indicator of deprivation. The likelihood of not accessing the vaccine increased with increasing deprivation of the residential census block. Moreover, women were 6% more likely than men not to get vaccinated, and foreign residents had triple the probability of Italians of not accessing the vaccine.

According to Our World in Data (ourworldindata.org), in December 2021, Italy was among the 10 European Countries with the highest share of people who completed the initial COVID-19 vaccination protocol (with a percentage of 74%), following Portugal, Malta, Spain, Denmark, Ireland, Iceland, and Belgium. The Italian figures were comparable with other countries in WHO regions, such as Canada, Australia, and New Zealand (Appendix A). The reported 74% differed from the 88% in our study population because we restricted the analysis to the adult population.

The higher non-vaccination in women is coherent with reports from other countries [21]. Elliott and colleagues found a lower intent to vaccination in women than in men [18]. An international survey on willingness to receive the COVID-19 vaccination showed a higher propensity in men than women in Germany, France, Russia, and the US, but not in China, where men were more hesitant than women [16].

Socioeconomic disparities in vaccination acceptance have been reported in Israel using area-based indicators [22] and individual indicators of socioeconomic position [17]. Caspi and colleagues found lower vaccination percentages in areas characterized by low socioeconomic status and high active disease burden. At the same time, Green and colleagues reported that higher education was associated with less vaccine hesitancy. The highly educated were found to be more favorable to COVID-19 vaccination than the low educated in the US [23]. Racial disparities have been reported in 2009 H1N1 vaccination in the US [24], raising concerns for the current pandemic. In fact, in the US, specifically in California, the delivery of COVID-19 vaccinations to black individuals is significantly behind their non-Hispanic white counterparts [25]. Elliott and colleagues found a lower intent to get vaccinated in black than white and Asians, in essential workers, and the most vulnerable groups of the population [18]. A study conducted in Wales, UK, using administrative data, with a methodology similar to this study, found that in April 2021, subjects living in a more deprived area or belonging to an ethnic group other than white were less likely to be vaccinated [26].

This study had strengths and limitations. It was a population-based study on more than three million subjects with individual educational levels. The administrative data pictured the actual situation on the 22 December 2021, regardless of the willingness to get a vaccine. The study population was restricted to those with the educational level available. The characteristics of subjects not included for missing information on education could lead to an underestimation of the inverse association between socioeconomic position and absence of vaccination. Moreover, since the population of the Lazio Region lives mainly in the Metropolitan Area of Rome, these results can be compared with other metropolitan areas, with a high proportion of well-off residents.

Several factors can influence the decision to get vaccinated, including unclear and unreliable COVID-19 vaccine information, uncertainty about scheduling, medical mistrust, costs of vaccination, and fears of politicization or pharmaceutical influence [27]. In Italy, as in the UK, all these factors might have a role, except for the costs, because a universal National Health Service provides health care. For foreigners, there could be a problem of access to health services due to barriers given by the level of integration, of language, and knowledge of the Regional Health System. Moreover, the digital divide, associated with the socioeconomic gap, could have played a role in not accessing the vaccine program. The reasons underlying vaccination hesitancy or vaccine refusal can differ among socioeconomic groups of the population [28]. Hence, it is necessary to adopt multicomponent strategies tailored to specific under-vaccinated people and guided by their perspectives and needs [28,29]. Immunization program planners should consider an evidence-based approach that better describes these subpopulations and chooses specific communication methods [28]. Additionally, they should understand and tackle the underlying causes of vaccine hesitancy and refusal. 

In conclusion, these results show the need for actions to close the gaps in COVID-19 vaccine disparities. Otherwise, the burden of COVID-19 will increase for the underprivileged and amplify social inequalities. The results show the groups of the population who could be targeted by information and communication campaigns: low educated, residents in deprived areas, and foreign residents.

## Figures and Tables

**Figure 1 vaccines-10-00364-f001:**
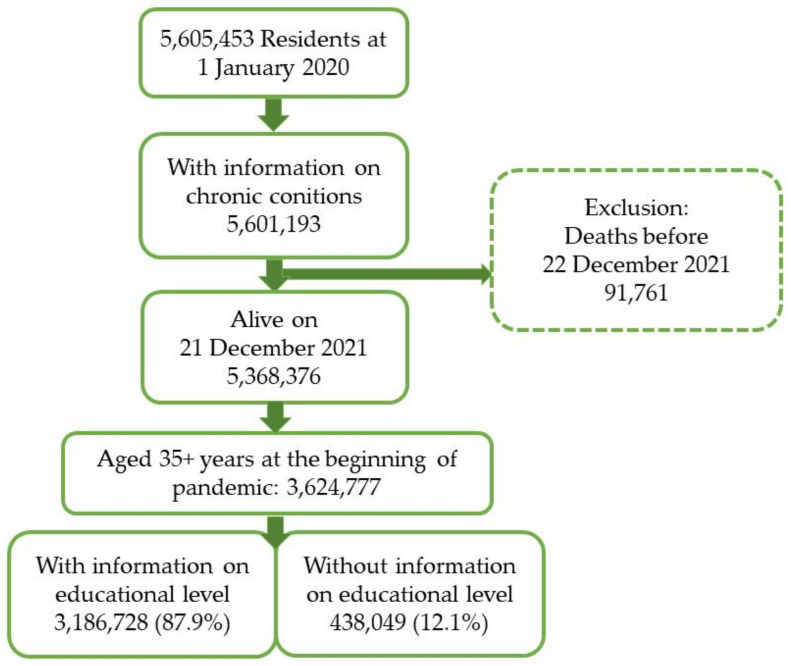
Flow chart of the study population selection.

**Table 1 vaccines-10-00364-t001:** Characteristics of the study population according to vaccination status on 22 December 2021.

Characteristic	N	%	% No Vaccination	% Partial Vaccination	% Completed Vaccination
Study population	3,186,728	100	10.3	1.5	88.1
Sex					
Males	1,467,434	46.0	10.0	1.5	88.4
Females	1,719,294	54.0	10.6	1.5	87.9
Age, mean (sd)	58.9 (14.3)		56.2 (14.8)	53.8 (13.9)	59.3 (14.2)
Place of birth					
Italy	2,931,917	92.0	9.0	1.5	89.5
Other Countries	254,811	8.0	25.4	2.1	72.5
Level of education ^1^					
High	576,382	18.1	8.9	1.4	89.7
Medium	1,245,099	39.1	11.0	1.6	87.4
Low	1,365,247	42.8	10.4	1.5	88.1
Deprivation quantile ^2^					
Q1 (low)	647,691	20.3	9.2	1.3	89.5
Q2	642,586	20.2	9.7	1.4	88.9
Q3	637,238	20.0	10.0	1.5	88.5
Q4	627,070	19.7	10.7	1.6	87.8
Q5 (high)	616,831	19.4	11.9	1.8	86.4
Number of chronic conditions					
0	1,597,725	50.1	13.0	1.9	85.1
1	738,096	23.2	8.5	1.3	90.2
2	403,689	12.7	7.1	1.1	91.8
3+	447,218	14.0	6.7	1.0	92.4

^1^ High = university degree, Medium = high school degree, Low = primary or junior high school degree. ^2^ The total is not 3,186,728 for 15,312 residential addresses were missing.

**Table 2 vaccines-10-00364-t002:** Characteristics associated with not getting vaccinated. Lazio Region, population aged 35+ years, 22 December 2021.

Characteristic	OR^1^	95%CI		OR^2^	95%CI	
Sex						
Males	1.000			1.000		
Females	1.086	1.078	1.094	1.063	1.055	1.071
Age (1 year increase)	0.985	0.985	0.985	0.996	0.996	0.997
Place of birth						
Italy	1.000			1.000		
Other Countries	3.187	3.155	3.218	2.968	2.938	2.998
Level of Education *						
High	1.000			1.000		
Medium	1.288	1.274	1.301	1.253	1.239	1.267
Low	1.414	1.399	1.430	1.391	1.375	1.407
Deprivation Quantile						
Q1 (Low)	1.000			1.000		
Q2	1.048	1.036	1.060	1.029	1.016	1.041
Q3	1.073	1.061	1.086	1.037	1.024	1.050
Q4	1.152	1.138	1.165	1.086	1.073	1.099
Q5 (High)	1.308	1.294	1.324	1.218	1.203	1.232
Number of Chronic Conditions						
0	1.000			1.000		
1	0.651	0.645	0.658	0.664	0.657	0.671
2	0.552	0.545	0.560	0.563	0.555	0.571
3+	0.531	0.523	0.538	0.537	0.529	0.544

* High = university degree, Medium = high school degree, Low = primary or junior high school degree. OR^1^ Base model: age and sex, + each variable one at a time. OR^2^ Fully-adjusted model (all the variables listed in the table).

## Data Availability

For privacy reasons, restrictions apply to the availability of the data. Individual data are accessible using strict rules on Lazio Region servers and cannot be exported. Aggregated data are available from the authors upon request.

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
