# Peer review of "Educational Inequalities in COVID-19 Vaccination: A Cross-Sectional Study of the Adult Population in the Lazio Region, Italy"

_vaccines, 2022, doi:10.3390/vaccines10030364_

Round 1
Reviewer 1 Report
Vaccine hesitancy is currently a major concern Worldwide. The present manuscript by Cesaroni et al. aimed at analyzing educational inequalities in COVID-19 vaccination. The topic is of high public health interest. The manuscript is well written, and the data are consistent with the study's conclusions. However, the discussion is weak and should be extended, at least, to the situation in all WHO health regions. The authors could also further discuss how education and social disparities foster vaccine hesitancy and propose concrete measures to close this gap.
Author Response
Reviewer
Vaccine hesitancy is currently a major concern Worldwide. The present manuscript by Cesaroni et al. aimed at analyzing educational inequalities in COVID-19 vaccination. The topic is of high public health interest. The manuscript is well written, and the data are consistent with the study's conclusions. However, the discussion is weak and should be extended, at least, to the situation in all WHO health regions.
Reply
We thank the reviewer for these comments. We agree the discussion was too concise, even for a brief report.
Following the suggestion, we extended the discussion. We added a paragraph on the vaccine situation in WHO health regions in lines 186-192
“According to Our World in Data (ourworldindata.org), in December 2021, Italy was among the 10 European Countries with the highest share of people who completed the initial COVID-19 vaccination protocol (with a percentage of 74%), following Portugal, Malta, Spain, Denmark, Ireland, Iceland, and Belgium. The Italian figures were comparable to other Countries in WHO regions, such as Canada, Australia, New Zealand (Figure S1). The reported 74% differed from the 88% in our study population because we restricted the analysis to the adult population”.
Moreover, we added the supplementary “Figure S1. Share of people with completed vaccination in December 2021 by Country” to facilitate the comparison between Countries.
In lines 236-238 we added: “Moreover, since the population of the Lazio Region lives mainly in the Metropolitan City of Rome, these results can be compared with other metropolitan areas, with a high pro-portion of well-off residents” to better describe the setting.
Reviewer
The authors could also further discuss how education and social disparities foster vaccine hesitancy and propose concrete measures to close this gap.
Reply
In lines 221-229 we added “Moreover, the digital divide, associated to the socioeconomic gap, could have played a role in not accessing the vaccine program. The reasons underlying vaccination hesitancy or vaccine refusal can differ among socioeconomic groups of the population [28]. Hence, it is necessary to adopt multicomponent strategies tailored to specific under-vaccinated people and guided by their perspectives and needs [28,29]. Immunization program planners should consider an evidence-based approach that better describes these subpopulations and choose specific communication methods [28]. Additionally, they should understand and tackle the underlying causes of vaccine hesitancy and refusal.”
Reviewer 2 Report
Introduction: authors quoted "Socioeconomic inequalities have been reported during the COVID-19 pandemic in 28 several Countries, raising solid concerns". This is mainly about inequalities among countries. While authors may need to explain more why in developed countries, socieoeconomic inequalities is found. This is due to improper promotion from government?
The research gap of this study "In Italy, the general population could get vaccinated, according to the age group and 35 starting from the elderly, during 2021. In December 2021, all the adult population that 36 wanted a COVID-19 vaccination had the possibility of being fully vaccinated. This study 37 aimed at evaluating the association between educational level and not getting vaccinated 38 on the 22nd December 2021." is relatively weak. Authors may need to explain about Vaccination programme in Italy, why
"efforts in reaching the low socioeconomic strata of the population" is not satisfied, etc.
Overall comment: Introduction must be improved, while other sections are clear.
Author Response
Reviewer
Introduction: authors quoted "Socioeconomic inequalities have been reported during the COVID-19 pandemic in 28 several Countries, raising solid concerns". This is mainly about inequalities among countries. While authors may need to explain more why in developed countries, socieoeconomic inequalities is found. This is due to improper promotion from government?
Reply: Thank you for this comment. Since it is a brief report, we assumed that the introduction had to be very concise. We now rewrote it. We tried to describe the context and explained the vaccination campaign as follows (lines 32-65):
Although media and scientific editorials addressed the COVID-19 as a great leveler at the beginning of the pandemic, commentaries and studies on socioeconomic inequalities during the pandemic soon appeared from several Countries, raising solid concerns [1,2]. Even in High-Income Countries, compared to the advantaged groups of the population, the disadvantaged have a higher risk of SARS-CoV-2 infection [3], higher hospitalizations for COVID-19 [4], increased access to intensive care unit [3], higher mortality rates [5–7], and lower probability of being tested for SARS-CoV-2 infection [3].
Social determinants of health have been deeply investigated in Europe and are pre-sent in all Countries [8]. In the Lazio Region, despite a universal Regional Health Service that aims to provide health care for all, socioeconomic inequalities in health and access to health care have been reported extensively [9-14]. Differences between social positions have been reported in life expectancy [9], in all-cause and cause-specific mortality [10], in the incidence and prognosis of diseases [11], in hospitalizations [12], in adherence to evidence-based drug therapies [13]. In addition, during 2020 educational inequalities in SARS-CoV-2 incidence and mortality within 30 days from the onset of infection have been analyzed [14], showing strong disparities in the last term of the year. In 2020, the vaccines against COVID-19 were not available yet, and the vaccination campaign started on the end of December 2020. Therefore, a comprehensive vaccination against COVID-19 could help in reduce socioeconomic inequalities. However, several studies investigated the characteristics associated with the willingness to receive the vaccination against COVID-19 [15–19], showing greater hesitancy in the disadvantaged groups of the population.
In Italy, vaccines were first offered to health care workers, then to the general population. The vaccine administration to the general population followed priority criteria based on individual vulnerability. The first to be vaccinated were the residents aged 80 years or more, disabled individuals, or subjects with severe comorbid conditions. The second were those aged 70-79 years. Then the 60-69-year-olds, followed by those aged less than 60 years with comorbid conditions or occupations characterized by high levels of exposure (teachers, police, and armed forces). Finally, vaccines were offered to younger groups of the population. Access to vaccination was possible through the general practitioner or in-dividual booking on the Regional Health Service website. the general population could get vaccinated, according to the age group and starting from the elderly, during 2021. In December 2021, all the adult population that wanted a COVID-19 vaccination had the possibility of being fully vaccinated. This The present study aimed at evaluating the association between educational level and not getting vaccinated on the 22nd December 2021 and at identifying specific targets for vaccination campaigns.
Reviewer
The research gap of this study "In Italy, the general population could get vaccinated, according to the age group and 35 starting from the elderly, during 2021. In December 2021, all the adult population that 36 wanted a COVID-19 vaccination had the possibility of being fully vaccinated. This study 37 aimed at evaluating the association between educational level and not getting vaccinated 38 on the 22nd December 2021." is relatively weak. Authors may need to explain about Vaccination programme in Italy, why
"efforts in reaching the low socioeconomic strata of the population" is not satisfied, etc.
Reply: we better explained the vaccination campaign (lines 53-62).
Efforts in reaching all the population have been made, in fact the percentage of vaccinated was 88%. However, a 10% of the population did not receive the vaccine. What we meant with the sentence "efforts in reaching the low socioeconomic strata of the population" is that further efforts are needed. Now we added “further” in the abstract.
Reviewer
Overall comment: Introduction must be improved, while other sections are clear.
Reply: We think that following your suggestion the introduction improved. We also corrected several English errors. Thank you.
Reviewer 3 Report
The paper presented for review is a well-performed study based on a Lazio COVID-19 cohort. There are, in my opinion, several issues that need to be addressed before publication, however.
- The study has been conducted in the Lazio Region that has a population of about 5.5 million, 4.3 mln of inhabitants living in Metropilitan City of Rome (2.8 mln in Rome City itself). This has to be clearly stated in the discussion and abstract to make sure that the readers will understand the results can be compared with other metropolitan areas rather than with general population.
- Consequently – 88.1% of the fully vaccinated population needs to be discussed carefully. This is a percentage rarely reached until today in general populations.
- Since the level of education is the main factor investigated we need more attention to it in the “Discussion”: possible influence of several Roman universities, very high percentage of highly educated, how does it opposed data showing problems in rural areas etc.
- Technically – why the only those aged 35 years or more were selected? (not 21+ or 18+…)?
Author Response
Reviewer 3
The paper presented for review is a well-performed study based on a Lazio COVID-19 cohort. There are, in my opinion, several issues that need to be addressed before publication, however.
- The study has been conducted in the Lazio Region that has a population of about 5.5 million, 4.3 mln of inhabitants living in Metropilitan City of Rome (2.8 mln in Rome City itself). This has to be clearly stated in the discussion and abstract to make sure that the readers will understand the results can be compared with other metropolitan areas rather than with general population.
Reply. Thank you for your suggestions, we think that the revision process improved the manuscript substantially.
We stated in the abstract:
(Lines 16-17) The Lazio Region has 5.5 million residents, mostly distributed in the Metropolitan City of Rome (4.3 million inhabitants).
We clarified in the methods section:
(Lines 69-71): Located in Central Italy, Lazio has a population of about 5.5 million inhabitants. The population lives primarily in the Metropolitan Area of Rome (4.3 million inhabitants) and Rome, the Italian Capital City, with 2.8 million residents.
And in the discussion section we wrote:
(Lines 236-238) Moreover, since the population of the Lazio Region lives mainly in the Metropolitan City of Rome, these results can be compared with other metropolitan areas, with a high proportion of well-off residents.
- Consequently – 88.1% of the fully vaccinated population needs to be discussed carefully. This is a percentage rarely reached until today in general populations.
Reply. We added a paragraph in the discussion section on the percentage of vaccinated:
(Lines 179-180): Our population had a high vaccination rate (88%), but we found solid inequalities in the COVID-19 vaccination.
(Lines 186-192): According to Our World in Data (ourworldindata.org), in December 2021, Italy was among the 10 European Countries with the highest share of people who completed the initial COVID-19 vaccination protocol (with a percentage of 74%), following Portugal, Malta, Spain, Denmark, Ireland, Iceland, and Belgium. The Italian figures were comparable to other Countries in WHO regions, such as Canada, Australia, New Zealand (Figure S1). The reported 74% differed from the 88% in our study population because we restricted the analysis to the adult population.
We added Figure S1 in supplemental material showing the share of people with completed vaccination in December 2021 by Country to help the reader in comparisons with other settings.
- Since the level of education is the main factor investigated we need more attention to it in the “Discussion”: possible influence of several Roman universities, very high percentage of highly educated, how does it opposed data showing problems in rural areas etc.
Reply. We specified in the limitations:
(Lines 236-238) Moreover, since the population of the Lazio Region lives mainly in the Metropolitan City of Rome, these results can be compared with other metropolitan areas, with a high proportion of well-off residents.
- Technically – why the only those aged 35 years or more were selected? (not 21+ or 18+…)?
Reply. Substantially, the choice was dictated by the availability of education information. It was retrieved by the 2011 census of the population, so we needed to select who had ended the educational path in 2011.
We better described census data:
(Lines 94-98): Since the last census of the population was conducted in 2011, the information on the in-dividual level of education was available for those living in the Lazio Region on the census reference day (9/10/2011) and can be considered valid for those who at the census reference day had completed the educational process, i.e. 25 years of age for university degrees.
And better explained the cut-off of 35 years:
Lines 105-106: From the cohort DeteCOVID of all residents, we selected the subjects aged 35 years or more on 1/1/2020, who presumably had completed education at the 2011 census.
Round 2
Reviewer 2 Report
Authors have addressed my comments, I have no more comments on their revised manuscript.